# Modulating Nitric Oxide Dioxygenase and Nitrite Reductase of Cytoglobin through Point Mutations

**DOI:** 10.3390/antiox11091816

**Published:** 2022-09-15

**Authors:** John Ukeri, Michael T. Wilson, Brandon J. Reeder

**Affiliations:** School of Life Sciences, University of Essex, Wivenhoe Park, Colchester CO4 3SQ, UK

**Keywords:** cytoglobin, nitric oxide, nitrite reductase, nitric oxide dioxygenase

## Abstract

Cytoglobin is a hexacoordinate hemoglobin with physiological roles that are not clearly understood. Previously proposed physiological functions include nitric oxide regulation, oxygen sensing, or/and protection against oxidative stress under hypoxic/ischemic conditions. Like many globins, cytoglobin rapidly consumes nitric oxide under normoxic conditions. Under hypoxia, cytoglobin generates nitric oxide, which is strongly modulated by the oxidation state of the cysteines. This gives a plausible role for this biochemistry in controlling nitric oxide homeostasis. Mutations to control specific properties of hemoglobin and myoglobin, including nitric oxide binding/scavenging and the nitrite reductase activity of various globins, have been reported. We have mapped these key mutations onto cytoglobin, which represents the E7 distal ligand, B2/E9 disulfide, and B10 heme pocket residues, and examined the nitric oxide binding, nitric oxide dioxygenase activity, and nitrite reductase activity. The Leu46Trp mutation decreases the nitric oxide dioxygenase activity > 10,000-fold over wild type, an effect 1000 times greater than similar mutations with other globins. By understanding how particular mutations can affect specific reactivities, these mutations may be used to target specific cytoglobin activities in cell or animal models to help understand the precise role(s) of cytoglobin under physiological and pathophysiological conditions.

## 1. Introduction

Cytoglobin (Cygb) is a low expression hexacoordinate globin found in all tissues of vertebrates [1]. Despite effort to assign a biological role to Cygb, the function of this protein is not clearly understood [2]. However, the proposed physiological roles of Cygb include scavenging nitric oxide (NO) [3,4], scavenging of reactive oxygen species [5,6], lipid peroxidation [7,8,9], and nitrite reductase activity (NiR) [4,10]. Observations that Cygb is upregulated in response to cellular hypoxia/ischemia [11,12,13] infer a potential role in either oxygen sensing (i.e., O_2_ homeostasis) or NO homeostasis. It has also been proposed that Cygb functions as a regulator of intracellular O_2_ homeostasis by acting as a redox-sensitive anti-oxidative protein in hepatic stellate cells [14].

As a result of its NO scavenging capability, Cygb could provide cytoprotective support for NO-sensitive enzymes, such as those proposed for myoglobin (Mb) and neuroglobin (Ngb). Both Mb and Ngb can protect terminal oxidases in the electron transport chain [15,16], and regulate the blood flow to hypoxic tissues in the vasculature by vasodilation through NO production [3,17]. Cygb has been reported to play a significant role in regulating vascular tone through NO scavenging in the vasculature. Downregulation of Cygb inhibits angiotensin-mediated hypertension and lengthens NO decay, leading to low blood pressure, highlighting the relevance of Cygb NO homeostasis in regulating vascular tone with implications for hypoxic/ischemic insult response [18,19]. The co-localization of Cygb with neuronal NO-synthase [20], as well as the implication of NO metabolism by Cygb in fibrogenesis [21], also links Cygb to NO metabolism [22]. The NiR-mediated NO generation by Cygb in anaerobic/hypoxic conditions, and oxygen-dependent NO consumption, suggest a potential regulatory role of Cygb in response to oxidative/nitrative stress [3,10,23]. The NiR and NO dioxygenase (NOD) activities performed by ferrous Cygb, oxidize the Cygb to the ferric oxidation state. Therefore, for these activities to be physiologically relevant, Cygb requires a rapid cellular reduction system to recycle the ferrous form. Cytochrome b5, cytochrome b5 reductase, NADH, and ascorbate all rapidly reduce ferric Cygb in vitro, potentially fitting the role of a cellular reduction system [22,24,25].

Cygb biochemistry can be influenced by the oxidation state of two surface-exposed cysteines that traverse two adjacent alpha helices; Cys38 on helix B (B2) and Cys83 on helix E (E9) [26,27,28]. As prepared, the recombinant protein is a mixture of a monomer with an intramolecular disulfide, a monomer with reduced disulfide, and a dimer with an intermolecular disulfide. The monomer with reduced cysteines and intermolecular dimer behave similarly to each other [29,30,31]. However, due to the proximity of E9 Cys83 to the E7 heme iron distal histidine, the presence of an intramolecular disulfide bond creates a compact structure changing the equilibrium of the distal histidine–heme iron ligation. This results in an increase in distal histidine off rate [29], making Cygb more pentacoordinate-like compared to the protein in the absence of this intramolecular disulfide bond, or as a dimer with intermolecular sulfide. The difference in structural configuration due to the redox state of the cysteines creates a cysteine redox state-dependent modulation of the binding or/and reactivity with exogenous ligands such as CO and NO [31]. The presence of the intramolecular disulfide bond switches the ligand migration pathway of CO [26], NiR activity [4], and lipid-induced changes in coordination [4,8].

Targeted mutagenesis of residues in the distal pocket of Mb, hemoglobin (Hb), and Ngb have been shown to specifically modulate either the NiR or/and NOD activities [32,33]. The NiR activity of Ngb was increased ~2000 times by mutation of the distal E7 histidine ligand [32]. A recent investigation into the effects of E7 mutation on Cygb NiR mutations showed varied effects ranging from 0.9 to 1100 M^−1^ s^−1^ [34]. Bimolecular ligands like NO have been shown to migrate to the distal coordination site of Hb through the rear of the heme pocket, to access the central iron. The distal histidine acts like a gate rotating out of place to enable biomolecular ligands to ligate to the central heme iron [35]. This mechanism can be impeded by removing the space at the rear of the heme pocket with large hydrophobic residues, such as phenylalanine or tryptophan, hampering access to the distal coordination site [33,35]. Mutation of neighboring residues of the distal histidine such as Leu(B10) was effective in decreasing the NOD activity of Hb and Mb by an order of magnitude [35,36]. These key amino acids that influence various molecular activities of globins, mapped onto the structure of Cygb, are shown in Figure 1.

Here we investigate the effect of mutation of the E7 distal histidine, B2, and E9 cysteines involved in the formation of the intramolecular disulfide and B10 leucine on the NO binding, NiR, and NOD activities of Cygb. Specific mutations resulted in significant modulation of NiR or/and NOD activity including a >10,000-fold decrease in NOD activity and a 6500-fold increase in NiR activity. The large effect observed on potential physiological reactions opens research avenues to explore the effects of specific mutations under physiological/pathological conditions in the cell line or in vivo research, and the responses to nitrative and oxidative stress.

## 2. Materials and Methods

### 2.1. Materials

Proli-NONOate was from Cayman chemicals, UK. Sodium phosphate (sodium dihydrogen orthophosphate and disodium hydrogen orthophosphate), sodium chloride, sodium tetraborate, sodium imidazole, kanamycin sulfate, and sodium chloride were purchased from Sigma-Aldrich, Poole, UK. Luria Bertani media was purchased from Melford laboratories Ltd., Ipswich, UK. Isopropyl B-D-1-thiogalactopyranoside and aminoleuvelic acid were obtained from Molekula Ltd., Darlington, UK.

### 2.2. Recombinant Human Cytoglobin Expression and Purification

Cygb was expressed in BL21DE3 *E. coli* cells (Merck, Gillingham, UK) cultured in Luria Bertani media and purified by immobilized metal affinity column, as previously described [4,8]. Dimeric protein with an intermolecular disulfide bond was removed by size exclusion chromatography (Sephadex G75), as previously described [31]. Following the concentration of the Cygb using a 5 kDa molecular weight cut-off spin filter (Merck, UK), the concentration of the Cygb was determined optically using the deoxyferrous protein (ε_428 nm_ = 165 mM^−1^ cm^−1^) [31]. For the mutants, the concentration was determined by measuring the heme content by reverse-phase HPLC, as previously described [31].

### 2.3. Site-Directed Mutagenesis of Human Cytoglobin

Cygb mutants were generated by site-directed mutagenesis using a modified Agilent quikchange II protocol on pET28a plasmid containing the human Cygb gene (Uniprot: Q8WWM9, gene accession number NM134268.5). Mutagenesis generated the following mutations and the following 5′ to 3′ primers: Leu46Phe forward GTGGGGGTGGCCATCTTCGTGAGGTTCTTTGTG; reverse CACAAAGAACCTCACGAAGATGGCCACCCCCAC. Leu46Trp forward GTGGGGGTGGCCATCTGGGTGAGGTTCTTTGTG reverse CACAAAGAACCTCACCCAGATGGCCACCCCCAC. His81Ala forward CCCCAGCTGCGGAAGGCCGCCTGCCGAGTCATG, reverse CATGACTCGGCAGGCGGCCTTCCGCAGCTGGGG. Cys38Ser forward CTCTATGCCAGCAGCGAGGACGTGGG, reverse: CCCACGTCCTCGCTGCTGGCATAGAG. Cys83Ser forward CGGAAGCACGCCAGCCGAGTCATGGG reverse CCCATGACTCGGCTGGCGTGCTTCCG. Mutations were confirmed by Sanger sequencing (Eurofins, UK).

### 2.4. Nitric Oxide Dioxygenase Activity

The NOD activity was initiated by reacting oxyferrous Cygb with nitric oxide at micromolar concentrations. Oxyferrous Cygb was produced by incubating 10 µM ferric Cygb with 5 mM sodium ascorbate for 15 min to get fully reduced protein. The complete reduction was monitored optically using an Agilent Cary 50 or Cary 5000 spectrophotometer. Sodium phosphate buffer (0.1 M, pH 7.4) was degassed using a tonometer linked to a vacuum pump and an argon gas source via a custom setup. The degassing was achieved by cycles of partial evacuation followed by an argon gas purge. The degassed buffer was transferred to 10 mL glass syringes anaerobically. Proli-NONOate was prepared by dissolving crystalline solid in 25 mM sodium hydroxide solution to a 40 mM proli-NONOate concentration. The concentration of proli-NONOate was determined by UV absorbance (ε_252 nm_ = 8400 M^−1^ cm^−1^). A micromolar concentration (10–400 µM) of proli-NONOate was transferred to the degassed buffer using a gastight glass Hamilton syringe. Upon the neutralization of the sodium hydroxide, Proli-NONOate disintegrates on the millisecond timescale to generate 1.8 molecules of NO in the buffer, but remains stable in deoxygenated condition until the reaction commences [38]. The NOD activity was initiated at 10 °C by 1:1 rapid mixing of 10 µM oxyferrous Cygb (5 µM after mixing) and NO (20–800 µM) using an Applied Photophysics SX-20 stopped-flow spectrophotometer.

### 2.5. Nitric Oxide Binding

The transition of deoxyferrous from hexacoordinated distal histidine-bound species to NO-bound species was monitored using an Applied Photophysics SX20 stopped-flow spectrophotometer. Degassed sodium phosphate buffer (0.1 M, pH 7.4) was prepared using custom degassing equipment and a tonometer using repeated cycles of partial evacuation with a vacuum pump followed by purging with argon gas. Deoxy Cygb (10 µM) was prepared through degassing via cycles of partial evacuation and argon gas purge with a custom setup that was linked to a vacuum pump and an argon gas supply. Partially deoxygenated protein was transferred to 10 mL glass syringes. Cygb was made ferrous and trace oxygen was removed by the addition of a minimum amount of dithionite. Proli-NONOate was transferred to 10 mL glass syringes with degassed buffer. Deoxyferrous Cygb (5 µM) after mixing) was rapidly combined (1:1) with NO (20–800 µM) by stopped flow at 20 °C.

### 2.6. Nitrite Reductase Activity

The NiR activity was monitored by reacting 5 µM Cygb with sodium nitrite (0–20 mM) in 0.1 M sodium phosphate buffer pH 7.4, in the presence of ~5 mM sodium dithionite, which was added preceding the reaction. Optical changes were measured using an Agilent 8453 diode array spectrophotometer fitted with multi-cell carriage and temperature control. The nitrite reductase activity of the His81Ala mutant was measured using the Applied Photophysics SX-20 stopped-flow spectrophotometer due to the more rapid activity. Kinetics was obtained from time courses at (416–429 nm) using the least squares methods with Microsoft Excel.

## 3. Results

### 3.1. Effect of Mutations on Nitric Oxide Binding to Cytoglobin

Optical changes in the binding NO to deoxyferrous wild-type (WT) Cygb are shown in Figure 2A. WT Cygb in the deoxyferrous form is hexacoordinate with a proximal and distal histidine ligated to the heme iron (His–Fe–His). This coordination results in the two sharp deoxyferrous α and β peaks in the visible spectrum, indicative of hexacoordinated globins. The optical changes in NO binding shift the Soret from 428 to 422 nm and are accompanied by a change in the visible region to a broad feature centered at 560 nm, typical of the NO-bound ferrous protein [4]. In Figure 2B the Cys38Ser/Cys83Ser (B2/E9) deoxyferrous spectrum is essentially identical to the WT protein with the NO-bound spectrum similar to the WT but with a slightly more prominent peak at 560 nm and a more prominent shoulder at ~540 nm. It is likely that the more prominent peak at 560 nm is a subpopulation that retains some coordination to the distal histidine, or that the geometry of the NO binding is different to that observed with pentacoordinate globins. The His81Ala mutation creates a typically pentacoordinate deoxyferrous spectrum in Figure 2C, as expected with this mutation, with a single visible peak at 560 nm, similar to deoxyferrous Hb and Mb. The NO-bound species is quite different compared with the other mutations and WT protein, with peaks at 530 and 560 nm with no 560 nm band. This is very similar to NO-bound ferrous Mb [39]. The optical characteristics of the Leu46Phe (B10, Figure 2D) mutant exhibit an identical deoxyferrous and ferrous NO spectrum to the WT protein. In Figure 2E, the Leu46Trp mutation changes the deoxyferrous spectrum so that the two α and β peaks in the visible region are less prominent and more like a pentacoordinate protein such as Mb, and the 560 nm band in the NO-bound species is much less prominent.

The kinetics of NO binding to deoxyferrrous protein as a function of NO concentration are shown in Figure 3 and summarized in Table 1. NO binding to WT cytoglobin was a biphasic process, but was essentially monophasic with all the mutants. Previous studies examining the binding of CO to Cygb also show a biphasic binding process [29,31]. This was identified to result from a heterogenous population of monomeric species, with the majority of the protein having an intramolecular disulfide bond (rapid binding) and a small subpopulation of a monomer with free sulfhydryl’s (slow binding) [29]. The biphasic binding of NO to WT Cygb can be assigned to this heterogenous population of cysteine oxidation states. With our preparation, the percentage of slow NO binding, assigned to the monomer with free sulfhydryl, was ~20–25% based on optical change (Figure 3B). The Cys38Ser/Cys83Ser mutation shows a monophasic NO binding with a slow NO binding rate constant (~0.2 s^−1^), supporting the proposition that the slow rate of NO binding, like that observed with CO binding, is due to the minor subpopulation of WT protein with a reduced cysteine oxidation state. For the WT Cygb, the rapid kinetics of NO binding assigned to the monomer with an intramolecular disulfide bond is essentially concentration-independent over the concentrations of NO used. Additionally, the Leu46Phe and Cys38Ser/Cys83Ser are also NO concentration-independent. The k_obs_ of NO binding are 11.7 (±1.1), 13.3 (±3.0), and 0.29 (±0.06) s^−1^ for WT (fast), Leu46Phe, and Cys38Ser/Cys83Ser respectively. These binding rates appear to be controlled by the distal histidine off rate (k_-H_) as the rates are typical of the k_-H_ rates observed with the WT protein [29]. The Cys38Ser/Cys83Ser mutant NO binding rate is close to the previously reported k_-H_ rate constant of 0.17 s^−1^ for the monomer with free sulfhydryl [29]. The WT and Leu46Phe variants with an intramolecular disulfide bond exhibit more rapid NO binding, close to the CO binding rates previously reported for WT Cygb [31]. The Leu46Trp and His81Ala mutants show a concentration dependence that is linear with respect to the NO concentration (Figure 3). NO binding to the Leu46Trp and His81Ala mutants was rapid in comparison to the WT, and exhibited a second order rate constant of 4.02 × 10^5^ M^−1^ s^−1^ (±4.0 × 10^4^ M^−1^ s^−1^) and 7.00 × 10^6^ M^−1^ s^−1^ (±3.0 × 10^5^ M^−1^ s^−1^) for Leu46Trp and His81Ala respectively, with a high apparent NO disassociation of 476 s^−1^ with the His81Ala mutant.

### 3.2. Effect of Mutations on the Nitrite Reductase Activity of Cytoglobin

The reaction of deoxyferrous Cygb with nitrite under anoxic conditions using sodium dithionite generates NO. Due to the presence of excess dithionite, the end spectrum is NO-bound deoxyferrous. Under hypoxic conditions in the absence of dithionite, the end species are a mixture of ferric and NO-bound ferrous; however, the rate constants of the NiR activity of globins are not affected by the presence or absence of dithionite [40]. The NiR activity of monomeric Cygb with an intramolecular disulfide exhibits biphasic kinetics with fast and slow phases each fitting to an exponential function. As with NO binding, we assign the fast phase to the monomeric Cygb with intramolecular disulfide, and the slow phase to the dimeric protein or monomer with free sulhydryls, based on our previous examination of the effects of the cysteine oxidation states [4]. Here, the dimeric protein was removed by gel filtration leaving the monomeric protein with an intramolecular disulfide (~80%) with the remaining protein with free sulfhydryls, similar to that observed with NO binding.

The kinetics of the NiR activity of Cygb are shown in Figure 4 and summarized in Table 1. The WT monomer with intramolecular disulfide is shown as a function of nitrite concentration, and the second-order rate constant was 31.7 ± 2.1 M^−1^ s^−1^, essentially identical to that reported previously [4]. The Cys38Ser/Cys83Ser double mutant showed a NiR second-order rate constant of 0.17 ± 0.01 M^−1^ s^−1^_,_ slightly lower than the 0.63 reported for monomer with free sulfhydryl, but closer to the 0.23 M^−1^ s^−1^ reported for a double cysteine to arginine mutant [4]. The Leu46Phe and Leu46Trp mutations exhibited NiR rate constants of 8.54 ± 0.25 M^−1^ s^−1^ and 0.19 ± 0.01 M^−1^ s^−1^, respectively, showing that the phenylalanine and tryptophan mutations in the heme pocket slow NiR activity to different extents. Finally, the His81Ala mutation leads to an NiR of 1108 ± 38.0 M^−1^ s^−1^, identical to that recently reported [34]; this is 35-fold higher in the rate of nitrite consumption compared to the WT with intramolecular disulfide, and 6500-fold higher compared to the double cysteine mutant.

### 3.3. Effect of Mutations on Nitric Oxide Dioxygenase Activity of Cytoglobin

The reaction of NO with oxyferrous Cygb is shown in Figure 5, and rate constants are summarized in Table 1. Here, the NO reacts with the oxygen bound to Cygb to generate nitrate (via peroxynitrite) and ferric Cygb (Fe^2+^ = O_2_ + NO → Fe^3+^−ONO^−^ → Fe^3+^+ NO_3_^−^). The issue with monitoring this reaction is that it is typically very fast, with rate constants reported using other globins typically >1 × 10^8^ M^−1^ s^−1^ [39]. For the WT Cygb, optical changes are observed (Figure 5A). However, the spectrum immediately following NO addition to oxyferrous Cygb (~1.2 ms by stopped-flow measurement) is not that of the oxyferrous species but a mainly pentacoordinate ferric Cygb. Therefore, the optical changes observed are typical of the ferric protein relaxing from a pentacoordinate to a hexacoordinate state, with the NOD reaction being largely missed within the dead time of the stopped-flow spectrophotometer (<1.2 ms). The kinetics of this relaxation approaches 284 s^−1^, fitting to a hyperbola as a function of NO concentration (Figure 5B). The dependence on NO concentration of the rate constant observed in Figure 5 for the WT, Cys38Ser/Cys83Ser, and Leu46Phe, we attribute to the fact that at low NO concentrations the NOD reaction is not fully completed. Thus, at these concentrations, the kinetics are confounded by contributions from two processes, namely for the formation of the ferric and the binding of the distal histidine.

For the Cys38Ser/Cys83Ser and Leu46Phe mutations, the optical changes and kinetics were essentially identical to that of the WT protein (Figure 5C,E) with the reaction being largely complete within the dead time of the stopped-flow spectrophotometer. Again, the optical changes are typical of the ferric protein relaxing from a pentacoordinate to a hexacoordinate state which approaches the rates of ~427 s^−1^ and ~298 s^−1^ for the Cys38Ser/Cys83Ser double mutation and Leu46Phe mutant respectively (Figure 5D,F). However, with the Leu46Trp mutant, we observed a concentration-dependent reaction, with the initial stopped-flow spectrum being primarily that of oxyferrous Cygb (Figure 5G). The subsequent optical changes monitor the formation of the final ferric protein. The observed rate constants ranged from 1.8–18.5 s^−1^ (10–100 µM NO) giving a second-order rate constant of 2.1 × 10^3^ M^−1^ s^−1^. This value represents the true NOD rate constant for the protein. With the WT, Cys38Ser/Cys83Ser, and Leu46Phe variants, the kinetics are relatively [NO] independent and reflect the relaxation of the distal ligand.

## 4. Discussion

The mutation of key residues in the structure of Cygb introduced significant effects on specific enzymatic reactivities of Cygb. The NiR activity of Cygb was decreased by targeting the cysteines (Cys38Ser/Cys83Ser), which has been reported earlier (via (tris(2-carboxyethyl)phosphine) reduction or by Cys38Arg/Cys83Arg mutation) to modulate the distal histidine, affecting the binding of the external ligand to the distal pocket [4]. This was in agreement with our current work, with a ~180-fold decrease observed in the rate of reaction as a result of the cysteine mutation (Figure 4). We assign this effect to a decrease in the k_-H_ rate resulting from a more relaxed endogenous distal histidine [29].

Replacement of the distal histidine E7 endogenous ligand by a smaller ligand (His81Ala) led to rapid biochemical activity in Cygb, supporting the gating effect of distal histidine on hexacoordinate globin, and also supporting previous studies on various globins that show that this histidine modulates the binding of the external ligand to the distal site [32,34,41]. Removal of this histidine reduces steric hindrance leading to rapid ligand binding to the distal site; the distal ligand is known to modulate ligand entry to the distal coordinate site in hemoglobins [35].

The effect of mutation on ligand migration in the distal pocket of Mb and Hb have been studied by Olson et al. [36,42,43]. While the histidine gating mechanism primarily controls ligand entry to the active site of the protein, secondary pockets and alternative cavities also play an important role in ligand coordination to the central heme iron, and these sites also enable ligands to achieve the configuration needed to bond to the protein [43]. In these studies, attenuation of ligand entry was achieved using the Leu29Trp (B10) mutation in Mb and Hb, decreasing the NOD activity by an order of magnitude [33]. Similar results were achieved here with Cygb, albeit with a much greater effect, being four or five orders of magnitude. In addition, the mutation of the B10 amino acid to a phenylalanine in Mb and Hb gave similar results as tryptophan for NOD activity decrease, but not in this case with Cygb. The tryptophan, although offering more bulky steric hindrance to incoming NO compared to phenylalanine, nonetheless gives similar effects as phenylalanine with Mb and Hb. The Leu46Trp mutation appears to form a more pentacoordinate-like heme iron geometry than that of the Leu46Phe mutation (Figure 1), thus potentially affecting the NOD reactivity, as pentacoordinate globins also typically show high NOD activity. However, the NiR activity of the Leu46Trp mutant is not enhanced like that observed with the His81Ala mutation. An explanation could be that the position of the phenylalaine and tryptophan side chains in the Leu46 mutations of Cygb are not the same, and have different steric influences on the approach of NO to the oxyferrous iron for NOD activity. This in turn may have effects on NOD and NiR activities, perhaps via a gating-type mechanism for the substrate entry or product exiting the heme pocket, or influencing the NO to O_2_ orientation. However, this needs to be confirmed by structural studies.

In previous studies on the NOD activity of globins, it has often been reported that rates of NOD are of the order of a few hundred per second [44]. Here we also observed rates of a few hundred per second with most protein variants, which we attribute to the distal histidine on rate. However, in this case, this is not the true NOD activity but merely the settling of the heme pocket geometry. Although Hb and Mb are pentacoordinate, some changes in heme pocket geometry following the completion of the formation of nitrate cannot be ruled out. This may cast doubt on the interpretation of some results in the literature, particularly if using single wavelength analysis.

Previous work examined the impact of the B10 Leu29Phe and Leu29Trp mutations (equivalent positions to the Leu46Phe and Leu46Trp on Cygb) on ligand migration in Mb and Hb [33,42]. The Leu29Phe mutation had minimal impact on ligand entry to myoglobin in contrast with the Leu29Trp mutation, which markedly affected ligand migration. The reason for this was the increasing energy barrier making bond formation more difficult [43]. The NO binding data (Figure 3) shows that the Leu46Trp changes the NO binding, increasing the rate (but not to the same extent as His81Ala) and making the rate not dependent on the slow distal histidine off rate. This means that the distal histidine off rate is increased by the Leu46Trp mutation. However, the suppressed NOD activity of the Leu46Trp may not be a reflection of this change in histidine off rate, as the Cys mutations (or Cys reduction) also change the histidine off rate [29,31].

## 5. Conclusions and Future Directions

The mutations chosen for this study change the specific activities of Cygb that in the literature have been suggested to play physiological roles in vivo. Although more work needs to be done to examine the effect of these mutations on other potential functions, (e.g., O_2_ binding, peroxidase activity), this in vitro study provides a pathway to examine the effect of these mutations in cell lines. For example, it is believed that Cygb plays a role in NO homeostasis, and that this may be linked to the protection of certain cancers by Cygb, particularly hypoxic tumors [13]. Measuring the cytoprotective effect of upregulated (or in this case induced) mutant Cygb expression in cancer cell lines (whether the mutations enhance or suppress specific activities) will be informative for an understanding of the mechanism of the protective effect on cancer cells in vivo. The effect of changes in the NOD or NiR activity will evidently have an effect on the oxidation state of the protein in the cell, and hence the ability to reduce the Cygb, e.g., by NADH, cytochrome b5 reductase, and cytochrome b5 may be affected. In addition, potential changes in autoxidation through mutations would also need to be evaluated on the cellular oxidation state of the Cygb. Nonetheless, NO homeostasis also plays an important role in cancer pathology [45], and attenuating the NOD and NiR activity of Cygb by mutagenesis could serve as a strategy to develop a more complete understanding of the mechanisms involved in the cytoprotective effects of Cygb, and therefore to develop methods to target this protective mechanism to sensitize tumors to oxidative stress.

## Figures and Tables

**Figure 1 antioxidants-11-01816-f001:**
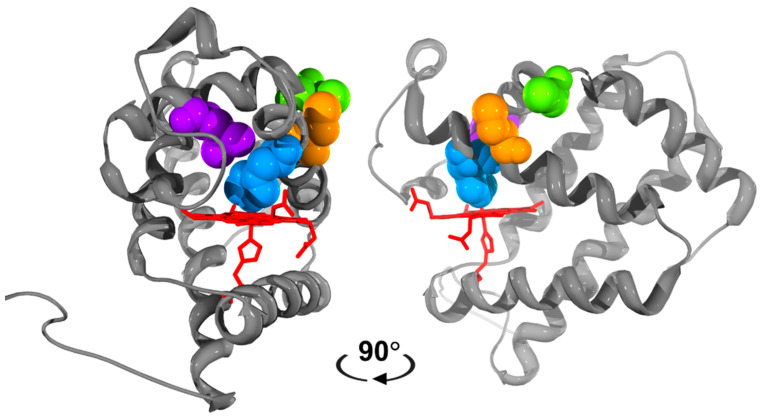
Structure of cytoglobin showing the key amino acids targeted for mutation. The heme and the F8 proximal histidine (red, stick) are shown with the heme edge on. The E7 histidine (H81) in blue spacefill, the E2 (Cys38) and E7 (Cys83) cysteines in green and orange, respectively, and the B10 Leu46 residue in purple. Structures are rotated 90°. Note that a structure with an intramolecular disulfide is not currently available. Swiss PDB viewer was used with raytracing by POVRAY for Windows v3.6. The structure was obtained from the RSCB protein databank (2DC3, [37]).

**Figure 2 antioxidants-11-01816-f002:**
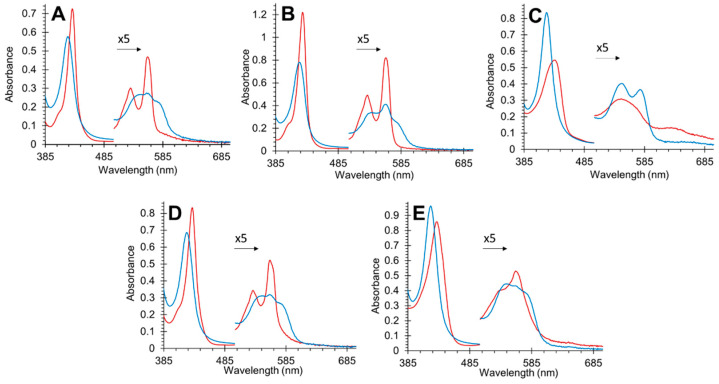
Optical spectra of deoxyferrous cytoglobin before (red) and after (blue) binding NO (720 µM) for wild type (**A**), Cys38Ser/Cys83Ser (**B**), His81Ala (**C**), Leu46Phe (**D**), and Leu46Trp (**E**). Visible region over 500 nm is multiplied by five for clarity.

**Figure 3 antioxidants-11-01816-f003:**
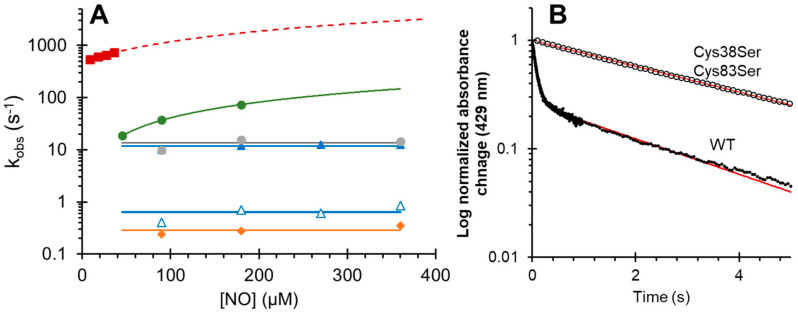
Rate constants for the nitric oxide binding to deoxyferrous cytoglobin as a function of NO concentration. (**A**) Blue closed triangles represent wild-type cytoglobin fast phase and blue open triangles represent wild-type cytoglobin slow phase, orange diamonds represent Cys38Ser/Cys83Ser double mutant, grey circles represent Leu46Phe mutation, green circles represent Leu46Trp mutation, and red squares represent His81Ala mutation. Leu49Trp and His81Ala mutations show a linear relationship as a function of NO concentration. Observed rate constant scale plotted on a logarithm scale for clarity (**B**) Logarithmic plot showing example time courses for wild type cytoglobin (closed circles) and Cys38Ser/Cys83Ser mutation (open circles). The biphasic time course of the wild-type protein is absent with the double mutant. NO concentration was 90 µM for both proteins; the red line represents fit to a double exponential function for the wild-type cytoglobin and a single exponential function for the double mutant.

**Figure 4 antioxidants-11-01816-f004:**
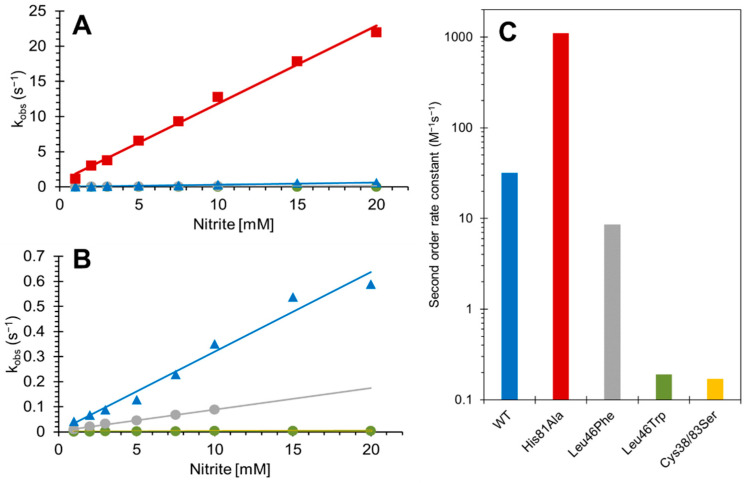
Effect of mutations on the rate constants of the nitrite reductase activity of cytoglobin. (**A**) Rate constants for the formation of ferrous NO from the reaction of Cygb and nitrite, and the effect of mutations on the reaction. (**B**) Expanded view of (**A**). (**C**) Second order rate constants of NiR activity of Cygb mutants (log scale). The color scheme is the same as that in Figure 2; Blue triangles represent wild−type cytoglobin, orange diamonds represent Cys38Ser/Cys83Ser double mutant, grey circles represent Leu46Phe mutation, green circles represent Leu46Trp mutation, and red squares represent His81Ala mutation.

**Figure 5 antioxidants-11-01816-f005:**
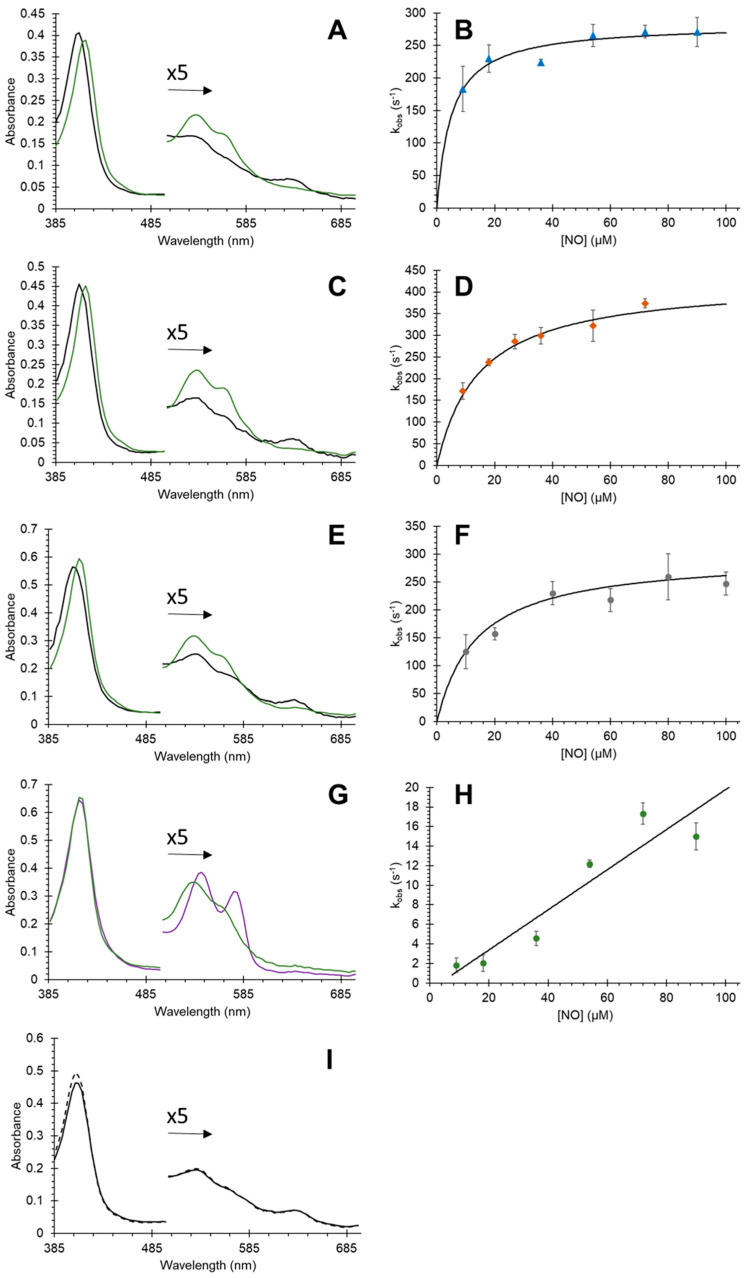
Effect of point mutations of nitric oxide dioxygenase activity of cytoglobin. (**A**), Change in optical spectra of WT Cygb following the addition of NO (20 µM) to oxyferrous protein (5 µM). The first recorded spectrum at ~1.2 ms (black line) showed ferric Cygb rather than oxyferrous Cygb. In addition, it is in the mainly pentacoordinate state. The final spectrum is that of the ferric protein in a hexacoordinate form (green line). (**B**) The kinetics of WT (**A**) as a function of [NO] shows slight concentration dependence. (**C**,**D**) Changes in spectra and kinetics for Cys38Ser/Cys83Ser mutation showing similar changes and activity to those of the WT. (**E**,**F**) Changes in spectra and kinetics for Leu46Phe mutation showing similar changes and activity to those of the WT and cysteine double mutant. (**G**,**H**) Changes in spectra and kinetics for Leu46Trp mutation showing that the initial spectrum is primarily still oxyferrous (purple line) with a significantly decreased rate of oxidation to the hexacoordinate ferric protein (green line) without exhibiting the pentacoordinate form. All reactions were at 20 °C in 100 mM sodium phosphate pH 7.4. (**I**) The His81Ala protein was ferric pentacoordinate immediately after mixing oxyferrous with NO (black solid line) and remained ferric pentacoordinate (dashed line 10 s after mixing). No measurable kinetics were observed for His81Ala, the reaction being complete <1 ms.

**Table 1 antioxidants-11-01816-t001:** NO binding, nitrite reductase, and nitric oxide dioxygenase activity of wild-type cytoglobin and mutants. NO binding is given as a first order (NO concentration independent) rate constant (s^−1^) or a second order (NO concentration independent) rate constant (M^−1^ s^−1^). ND represents not determined.

Mutation	NO Binding	NiR (M^−1^ s^−1^)	NOD (M^−1^ s^−1^)	NOD Spectra Relaxation (s^−1^)
Wild Type (intramolecular disulfide)	11.7 (s^−1^)	31.7	>1 × 10^8^	284
Cys38Ser/Cys83Ser	0.29 (s^−1^)	0.17	>1 × 10^8^	427
His81Ala	7.0 × 10^6^ (M^−1^ s^−1^)	1108	>1 × 10^8^	ND
Leu46Phe	13.3 (s^−1^)	8.54	>1 × 10^8^	298
Leu46Trp	4.0 × 10^5^ (M^−1^ s^−1^)	0.19	2.1 × 10^3^	ND

## Data Availability

All of the data is contained within the article.

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
