# Peer review of "Modulating Nitric Oxide Dioxygenase and Nitrite Reductase of Cytoglobin through Point Mutations"

_antioxidants, 2022, doi:10.3390/antiox11091816_

Round 1

Reviewer 1 Report

The paper by Ukeri et al. on the role of cytoglobin in the NO/O2 metabolism indeed is very interesting and it deserves publication on Antioxidants, but only after carrying out a series of modifications.

Two main points have to be completed by the authors.

First of all, NO dissociation rate constants have to be determined for WT and for all mutants directly (e.g., by ligand displacement) and not simply guessed on the basis of the NO concentration dependence. This will allow to establish the role of distal His, intra-molecular disulphide bond and mutated residues not only on the accessibility of ligands to the binding site but also on the escape pathway of the ligand.

A second point concerns the NO dioxygenase reaction and the Leu46Trp mutant behaviour, displaying the slowest rate, which seems to be in contradiction with its capability of increasing the amount of penta-coordinated cytoglobin. Actually, authors state correctly that the mechanism must be more complex than simply a hexa-to-penta-coordinate transition, but they should make an effort to deepen and clarify the possible scenario.

In addition, there are other minor points, where authors should intervene, namely:

(i)                I suppose that the very slow rate of NO binding, observed for the mutant Cys38Ser/Cys83Ser, is similar to the slower rate of WT, but, assuming this is true, this is never clearly stated and commented by the authors; on the other hand, if the rate is not very similar to the slower one of the WT, the authors should comment it.

(ii)              Dependence of rate constants on NO concentration (in Figs. 3A and 3B, as well as in Figs. 5B, 5D, 5F and 5H) and on Nitrite (in Figs. 4A  and 4B) should be better grouped in a single figure (different for the various reactions, obviously), employing a logarithmic ordinate scale, which should allow a better comparison.

Author Response

First of all, NO dissociation rate constants have to be determined for WT and for all mutants directly (e.g., by ligand displacement) and not simply guessed on the basis of the NO concentration dependence. This will allow to establish the role of distal His, intra-molecular disulphide bond and mutated residues not only on the accessibility of ligands to the binding site but also on the escape pathway of the ligand.

  • We appreciate that measurement of NO dissociation rates are of interest here. However, given that the affinity of Cygb for NO is very high, much higher than alternative ligands such as CO or O2, NO dissociation rate constant determination by ligand displacement is not a straightforward experiment. High concentrations of the ligand to displace the NO would be needed which is difficult to achieve with O2/CO, making the % NO displacement achievable very low. This is indeed an experiment worthy of further study and it maybe that other ways to measure the NO displacement (e.g. rapid sequestering of NO following dissociation) are worth investigating. However, the primary aim of this manuscript is to show which mutations could be used to study the potential physiological/pathophysiological activity of Cygb. We will certainly examine further aspects including the properties of NO dissociation for future publications, but we feel that the absence of this data would not substantially detract from the conclusions of this study.

A second point concerns the NO dioxygenase reaction and the Leu46Trp mutant behaviour, displaying the slowest rate, which seems to be in contradiction with its capability of increasing the amount of penta-coordinated cytoglobin. Actually, authors state correctly that the mechanism must be more complex than simply a hexa-to-penta-coordinate transition, but they should make an effort to deepen and clarify the possible scenario.

  • As the reaction of NO with Cygb is with the oxyferrous state, the deoxyferrous distal state of the globin may be of little relevance on NOD activity, unlike that of NiR activity. What may be of more consequence is the position of the displaced distal ligand when oxygen is bound and if it acts as a gate for NO entry into the heme pocket or nitrate expulsion. As with previous publications, the NOD activity appears to be primarily influenced by access of the NO to the back of the heme pocket. However, more telling is the lack of a relationship between pentacoordination of the Leu46 mutants and NiR activity, which appears to be the main influence with the His81Ala mutation. Thus an explanation for both observations could be that the mutation affects the position of the displaced distal ligand, maybe acting as a gating mechanism for entry/exit from the heme pocket. We have clarified this in the discussion.

In addition, there are other minor points, where authors should intervene, namely:

(i) I suppose that the very slow rate of NO binding, observed for the mutant Cys38Ser/Cys83Ser, is similar to the slower rate of WT, but, assuming this is true, this is never clearly stated and commented by the authors; on the other hand, if the rate is not very similar to the slower one of the WT, the authors should comment it.

  • The reviewer is correct that the slow rate of NO binding to WT protein, like that observed previously with CO binding, is due to a subpopulation of protein with free sulfhydryl’s. We have clarified this in section 3.1.

(ii) Dependence of rate constants on NO concentration (in Figs. 3A and 3B, as well as in Figs. 5B, 5D, 5F and 5H) and on Nitrite (in Figs. 4A and 4B) should be better grouped in a single figure (different for the various reactions, obviously), employing a logarithmic ordinate scale, which should allow a better comparison.

  • We have changed figure 3 to that of a log plot as suggested. However, for Figure 4 we feel that a split figure is acceptable as the main point of the figure is to show the linear relationship between the rate constant observed and the nitrite concentration. Figure 5, while more compact with combined data is complex and difficult to interpret. Thus we request that Figures 4 and 5 remain in their original format.

Reviewer 2 Report

Ukeri et al present a study of the effect of heme pocket mutations on the nitric oxide dioxygenase activity of cytoglobin. Several positions are studied including L46 (B10), His81 (E7) and the disulfide bond formed by the Cys38/Cys83 residues.

This is a well conducted study and the conclusions are well supported by the data. Overall the study is suitable for publication in its current form. I only have some minor questions/comments-

1. Did the authors quantify the ProliNONOate concentration based on NO release (e.g. by formation of methemoglobin or similar methods)? The methods section states that ProliNONOate releases 2 NO molecules per molecule; personally, I found the yield to be around 1.6-1.8 moles NO/mol ProliNONOate; similar results are reported elsewhere (Simons, M., ... & Cooper, C. E. (2018). Comparison of the oxidative reactivity of recombinant fetal and adult human hemoglobin: implications for the design of hemoglobin-based oxygen carriers. Bioscience reports, 38(4)) and yields below the expected value of 2 are common in other NONOates (Li, Q., & Lancaster Jr, J. R. (2009). Calibration of nitric oxide flux generation from diazeniumdiolate NO donors. Nitric Oxide, 21(1), 69-75.). Small differences in the real vs expected NO concentrations may bias the observed rates slightly.

2. The authors note this issue but certainly in the reaction of the mutants with NO (Figure 2) all the mutants show some deoxy species (except for His81Ala). Can this species be converted to the nitrosyl by addition of more NO? Does the final spectra in the nitrite reductase experiments show a 100% nitrosyl species or there is still some deoxy population that does not bind NO?

3. From the text it seems that the reported nitrite reductase rates only refer to the fast phase with an intramolecular disulfide. In the case of Leu46Trp and Leu46Phe, is there a slow phase detected or it would not be observed under the experimental conditions (<15% of dimer present)? It would be curious to see if the B10 mutations have an effect on the slow phases as well

4. A missing piece of the puzzle here is the autoxidation rate of the mutants, in particular for the Leu46Trp and Leu46Phe mutants. B10 mutations appear to be critical to autoxidation in myoglobin (Brantley, R. E., ... & Olson, J. S. (1993). The mechanism of autooxidation of myoglobin. Journal of Biological Chemistry, 268(10), 6995-7010) and neuroglobin Tejero, J., ... &Gladwin, M. T. (2015). Exploring the mechanisms of the reductase activity of neuroglobin by site-directed mutagenesis of the heme distal pocket. Biochemistry, 54(3), 722-733. Whereas the observed rates for NO dioxygenation seem much faster than those measured for the autoxidation of any cytoglobin or neuroglobin mutant, perhaps autoxidation is also much faster in the L46W mutant and contributing to the observed decay. Including the autoxidation rates for these two mutants would strengthen the authors’ conclusions.

5. Are there differences in the ascorbate reduction rates of the different mutants? 5mM ascorbate is not particularly high as cytoglobins do not get reduced very efficiently by ascorbate (Amdahl, M. B., … & Tejero, J. (2017). Efficient reduction of vertebrate cytoglobins by the cytochrome b 5/cytochrome b 5 reductase/NADH system. Biochemistry, 56(30), 3993-4004.) and considering the probable fast rates of autoxidation by Leu46Trp significant depletion of the ascorbate (and generation of ascorbate radicals) may occur.

6. Some values are reported with 4 or 5 significant digits which seems a bit excessive. Three significant digits are probably adequate in most cases.

Author Response

Reviewer 2:

Ukeri et al present a study of the effect of heme pocket mutations on the nitric oxide dioxygenase activity of cytoglobin. Several positions are studied including L46 (B10), His81 (E7) and the disulfide bond formed by the Cys38/Cys83 residues.

This is a well conducted study and the conclusions are well supported by the data. Overall the study is suitable for publication in its current form. I only have some minor questions/comments-

  1. Did the authors quantify the ProliNONOate concentration based on NO release (e.g. by formation of methemoglobin or similar methods)? The methods section states that ProliNONOate releases 2 NO molecules per molecule; personally, I found the yield to be around 1.6-1.8 moles NO/mol ProliNONOate; similar results are reported elsewhere (Simons, M., ... & Cooper, C. E. (2018). Comparison of the oxidative reactivity of recombinant fetal and adult human hemoglobin: implications for the design of hemoglobin-based oxygen carriers. Bioscience reports, 38(4)) and yields below the expected value of 2 are common in other NONOates (Li, Q., & Lancaster Jr, J. R. (2009). Calibration of nitric oxide flux generation from diazeniumdiolate NO donors. Nitric Oxide, 21(1), 69-75). Small differences in the real vs expected NO concentrations may bias the observed rates slightly.
  • The reviewer is indeed correct in that the ratio of NO to proli-NONOate is experimentally 1.8:1 instead of the theoretical 2:1 and we thank the reviewer for correcting this. Data and text have been adjusted to compensate in Figures 3 and 5. Only the NO concentration dependent second order rate constant values were marginally affected.
  1. The authors note this issue but certainly in the reaction of the mutants with NO (Figure 2) all the mutants show some deoxy species (except for His81Ala). Can this species be converted to the nitrosyl by addition of more NO? Does the final spectra in the nitrite reductase experiments show a 100% nitrosyl species or there is still some deoxy population that does not bind NO?
  • The final spectra with the NiR experiments all show that the end spectrum to be identical to that of the nitrosyl complex and increasing NO does not alter the end spectrum. In addition, other publications also show similar end spectra for WT Cygb (e.g. ref 10). Given that NO binding to the WT protein is high affinity, the optical properties could be assigned to a difference in NO binding geometry rather than a competition between the NO binding and distal histidine. However, as no crystal structures are currently available as the NO complex, we cannot be certain of this hypothesis. The text has been modified to reflect this in section 3.1.
  1. From the text it seems that the reported nitrite reductase rates only refer to the fast phase with an intramolecular disulfide. In the case of Leu46Trp and Leu46Phe, is there a slow phase detected or it would not be observed under the experimental conditions (<15% of dimer present)? It would be curious to see if the B10 mutations have an effect on the slow phases as well
  • There was no dimer present as the dimer was removed by SEC prior to analysis, the <15 % amplitude resulted from monomer with free sulfhydryl. We have previously reported that reduction of the disulfide (by TCEP) removes that fast phase and increases the amplitude of the slow phase (ref 31). This has been clarified in the methods. There was no slow phase detected with the B10 mutants, implying that the effect of the disulfide was either obscured or diminished with the B10 mutations.
  1. A missing piece of the puzzle here is the autoxidation rate of the mutants, in particular for the Leu46Trp and Leu46Phe mutants. B10 mutations appear to be critical to autoxidation in myoglobin (Brantley, R. E., ... & Olson, J. S. (1993). The mechanism of autooxidation of myoglobin. Journal of Biological Chemistry, 268(10), 6995-7010) and neuroglobin Tejero, J., ... &Gladwin, M. T. (2015). Exploring the mechanisms of the reductase activity of neuroglobin by site-directed mutagenesis of the heme distal pocket. Biochemistry, 54(3), 722-733. Whereas the observed rates for NO dioxygenation seem much faster than those measured for the autoxidation of any cytoglobin or neuroglobin mutant, perhaps autoxidation is also much faster in the L46W mutant and contributing to the observed decay. Including the autoxidation rates for these two mutants would strengthen the authors’ conclusions.

We complexly agree with the reviewer that the autoxidation rates would indeed be valuable when discussing the NOD activity. However, at present we do not have the mutants immediately available and thus we are unable to comply with the reviewers reasonable request. We have, however, added a sentence that recognised that autoxidation rates for these mutants are  important for a full understanding of their NOD activity.

  1. Are there differences in the ascorbate reduction rates of the different mutants? 5mM ascorbate is not particularly high as cytoglobins do not get reduced very efficiently by ascorbate (Amdahl, M. B., … & Tejero, J. (2017). Efficient reduction of vertebrate cytoglobins by the cytochrome b 5/cytochrome b 5 reductase/NADH system. Biochemistry, 56(30), 3993-4004.) and considering the probable fast rates of autoxidation by Leu46Trp significant depletion of the ascorbate (and generation of ascorbate radicals) may occur.
  • In our hands ascorbate reduction of Cygb and its mutants were very rapid. Much more so that other globins like Mb or Hb. 5-10 minutes was all that was needed to ensure essentially complete reduction with 5 mM ascorbate. This was assessed based on comparison of the spectrum to authentic oxyferrous species created by dithionite reduction and size exclusion chromatography. We wished to avoid the dithionite method as it partially reduced the disulfide over a period of tens of minutes. The reviewer is correct that a cytb5/b5 reductase system provides an alternate method to generate oxyferrous protein. However, this system would cause complications for NOD activity interpretation due to the presence of other heme proteins. Given the timescales of the stopped-flow experiment (milliseconds to seconds), the ascorbate reduction method was the most simplistic system to use.
  1. Some values are reported with 4 or 5 significant digits which seems a bit excessive. Three significant digits are probably adequate in most cases.
  • Noted and adjusted.

Reviewer 3 Report

This is a very interesting paper that uses site directed mutagenesis to investigate the role of key residues (identified by analogy with other globins) on the nitric oxide metabolising/reductase activity of cytoglobin. Overall the results are of great interest to researchers in the field of cytoglobin biochemistry as they not also shed direct light on the possible biochemical function of this elusive protein but also generate new hypotheses for cell biologists to investigate in future studies to link these structural changes to cellular function and biological properties of cytoglobin.

The manuscript is very well written and clearly presented and the data presented well, all the experimental approaches are appropriate and well controlled and the data presented supports the conclusions made by the authors. Overall I recommend publication of this manuscript as it stands

Author Response

We thank the reviewer for their kind comments.

Reviewer 4 Report

In this manuscript, Ukeri et al investigate one functional aspect of cytoglobin, a newly discovered heme protein, through point mutations. Despite being discovered 15 years ago, Cygb has still many controversial aspects that render it difficult to frame its function in the cell, thus this piece of work is useful to better understand the related functions. The work described in this manuscript has been obtained using fast-kinetic methods, appears original and adheres to good principles.

My main concern may be a question rather than a concern. I understand that Cygb occurs as a mixture of monomers and dimers, with dimers with higher O2 affinity than monomers and more prevalent in an oxidative environment. This variable seems to be efficiently ruled out by the Authors who performed their tests in the presence of excess ascorbate. However, in analogy with the dimer-tetramer shift in human Hb (R-like, high O2 affinity conformation for dimers vs T-like, low O2 affinity for tetramers), I wonder if the absolute concentration of the protein may influence the equilibrium between the two states. As the stopped flow tests were run at Cygb concentration of 5 microM, how can this choice affect the equilibrium? I understand that you may have chosen this concentration because close to the in vivo situation (but temperature, 20°C, is already far from the in vivo one), but perhaps it might be useful to run tests at varying Cygb concentrations to assess the possible effect of protein concentration on the distribution of the two subpopulations (monomers vs. dimers) at the same levels of the other variables.  

Possibly linked to the previous concern, in Line 187 it is not clear to me what is meant for “biphasic process”. Please report a representative kinetics in a log-graph to highlight this biphasic process. Figure 3A should report both for the slow and the fast components of the NO-dependent k values. As I explained above, I wonder, in analogy with the Hb binding to NO, that this might be due to the presence of varying amounts of the two subpopulations whose distribution may depend not only on the oxidative imbalance but also on the protein concentration. 

Still linked to my concern above, in Para 2.2, you should specify if the Cygb protein concentration in the affinity column eluate required adjustment or not. How was the protein concentration determined and adjusted to 5 microM? What is the variability of the protein concentration in the various runs?

A final major concern regards the second part of the conclusion that is rather speculative. I suggest moving it to the discussion under a separate paragraph labeled, say, “impact of these findings on the behavior of cancer cells”. In this context, avoid the term “protective” that may lead to misinterpretations: being protective for cancer cells is not protective for the cancer patient. 

A paragraph with the limits of the study will be highly appreciated. 

Minor

Line 9-10, syntax problem.

Line 102, the source for CO has been cited but in this manuscript there is no data obtained with it (too bad, it could be very informative).

Line 126, are the Authors sure that excess ascorbate is without influence on the measured variables? 

Line 171, perhaps traces of ferric Cygb may also affect position and height of the peak at 560 nm.

Line 200, how can the Authors state that “These binding rates appear to be controlled by the distal histidine off rate (k-H)”?

Line 221, “under hypoxic conditions”, in the presence of traces of dithionite, hypoxia must be replaced by ”anoxia”. 

Line 224, this may be a weak point. In the presence of oxygen traces, NO may be converted to ONO- that oxidizes ferrous to ferric Cygb. Please clarify.

Line 227, curve fitting on double exponentials might enable determing the distribution of the subpopulations. Were the spectra constant over time even some time after the stopped-flow kinetic determination? If no, there may have been some slow conversion of ferrous to ferric Cygb.

Figure 3B, I see only one green circle.

Table 1, what is the difference between ND and leaving a blank cell?

Author Response

My main concern may be a question rather than a concern. I understand that Cygb occurs as a mixture of monomers and dimers, with dimers with higher O2 affinity than monomers and more prevalent in an oxidative environment. This variable seems to be efficiently ruled out by the Authors who performed their tests in the presence of excess ascorbate. However, in analogy with the dimer-tetramer shift in human Hb (R-like, high O2 affinity conformation for dimers vs T-like, low O2 affinity for tetramers), I wonder if the absolute concentration of the protein may influence the equilibrium between the two states. As the stopped flow tests were run at Cygb concentration of 5 microM, how can this choice affect the equilibrium? I understand that you may have chosen this concentration because close to the in vivo situation (but temperature, 20°C, is already far from the in vivo one), but perhaps it might be useful to run tests at varying Cygb concentrations to assess the possible effect of protein concentration on the distribution of the two subpopulations (monomers vs. dimers) at the same levels of the other variables.  

  • We removed dimeric protein with an intermolecular disulfide bond by size exclusion chromatography (we have clarified this in the methods). Therefore any dimeric protein during the experiment would be from transient interactions. The allosteric equilibrium of such a transient complex would in the sub-millisecond timescale assuming a similar mechanism as observed in hemoglobin. Under the conditions of the experiments (i.e. in the absence of a redox agent capable of reducing and re-oxidizing the cysteines), re-establishment of an equilibrium between a covalent dimer would be far outside the timeframe of the experiment.

Possibly linked to the previous concern, in Line 187 it is not clear to me what is meant for “biphasic process”. Please report a representative kinetics in a log-graph to highlight this biphasic process. Figure 3A should report both for the slow and the fast components of the NO-dependent k values. As I explained above, I wonder, in analogy with the Hb binding to NO, that this might be due to the presence of varying amounts of the two subpopulations whose distribution may depend not only on the oxidative imbalance but also on the protein concentration. 

  • We have previously determined that the bi-phasic nature of the WT resulted from a mixture with the majority of the protein in the monomer (S-S) state (i.e. with an intermolecular disulfide) exhibiting a fast rate of reaction and a subpopulation of monomer (S-H) (i.e. monomer with free sulfhydryl’s) showing a slow rate of reaction (ref 31). We also previously incubated this mixture with TCEP to reduce the disulfide. Under these conditions the fast rate was virtually eliminated (that assigned to the monomer with intramolecular disulfide), with the majority of the reaction slow, resulting from monomer with free sulfhydryl. As mentioned above, we have no evidence that the kinetics are influenced by dimeric interactions. As suggested we have replaced the figure with a log plot - Figure 3B and added the rate constants for slow component of the WT protein.

Still linked to my concern above, in Para 2.2, you should specify if the Cygb protein concentration in the affinity column eluate required adjustment or not. How was the protein concentration determined and adjusted to 5 microM? What is the variability of the protein concentration in the various runs?

  • Method for concentrating protein and protein concentration determination have been clarified in section 2.2. Variation between concentrations of protein between experiments was minimal as judged by absorbance.

A final major concern regards the second part of the conclusion that is rather speculative. I suggest moving it to the discussion under a separate paragraph labeled, say, “impact of these findings on the behavior of cancer cells”. In this context, avoid the term “protective” that may lead to misinterpretations: being protective for cancer cells is not protective for the cancer patient. 

  • Text has been modified to clarify that these mutagenesis methods could be used to understand the mechanism of the cytoprotective effect of Cygb and hence develop methods to target this. We feel that this is a good position to finish the text to point to future directions, however, we agree that this is not a conclusion and have changed the title of the section to conclusions and future directions. We hope this is acceptable.

A paragraph with the limits of the study will be highly appreciated. 

  • We have emphasised the limitations of the study and what future research needs to be conducted in the Conclusions and future directions section.

Minor

Line 9-10, syntax problem. Text adjusted.

Line 102, the source for CO has been cited but in this manuscript there is no data obtained with it (too bad, it could be very informative). We have removed this citation.

Line 126, are the Authors sure that excess ascorbate is without influence on the measured variables? 

To our knowledge ascorbate did not interfere with experiments. Alternate use of dithionite (+ SEC) to generate oxyferrous Cygb resulted in a slow reduction of the disulfide.

Line 171, perhaps traces of ferric Cygb may also affect position and height of the peak at 560 nm.

Minimum amounts of sodium dithionite were added to the Cygb to ensure no ferric Cygb was present and minimal disulfide reduction in the timeframe of the experiment.

Line 200, how can the Authors state that “These binding rates appear to be controlled by the distal histidine off rate (k-H)”?

The numbers are essentially identical to those previously reported for k-H. A reference has been added to emphasise this.

Line 221, “under hypoxic conditions”, in the presence of traces of dithionite, hypoxia must be replaced by ”anoxia”. Changed.

Line 224, this may be a weak point. In the presence of oxygen traces, NO may be converted to ONO- that oxidizes ferrous to ferric Cygb. Please clarify.

  • With our system there is no oxygen due to the presence of dithionite, so a reaction with NO and O2 cannot occur allowing us to measure the pure NiR rate constants. In the presence of trace oxygen this is certainly a possibility to form ONO-. In vivo it will be apparent that the majority of the end product will be ferric cytoglobin due to the absence of dithionite. As described in section 3.2 (now line 247), the experiment here used dithionite to simplify the measurement of the rate of reaction without the need to use an anaerobic chamber. Previous studies (ref 40) have shown that the rate of reaction is not effected by the reaction done anaerobically in the absence or presence of dithionite, despite the different endpoint species.

Line 227, curve fitting on double exponentials might enable determing the distribution of the subpopulations. Were the spectra constant over time even some time after the stopped-flow kinetic determination? If no, there may have been some slow conversion of ferrous to ferric Cygb.

  • As mentioned above and at the start of the discussion, we have previously reported that reduction of the disulfide by TCEP (tris(2-carboxyethyl)phosphine) removes that fast phase and increases the amplitude of the slow phase (ref 31), identifying the slow phase as that of monomer with reduced cysteines. Assuming identical absorption coefficients, the slow phase for the WT protein was ~20-25% of the amplitude change. This was variable depending on the recombinant expression batch due to the different levels of oxidized and reduced cysteine in the monomer, but did not change when using the same protein batch.

Over the time course of the experiments this did not change, although in the presence of excess dithionite we have previously noted a change in CO binding resulting from the slow reduction of the cysteines over the course of several hours (ref 31). Thus in the experiments we either used minimum amounts of dithionite (for optical spectra) or dithionite was added immediately prior to reaction (NiR activity) in order to minimize cysteine reduction over the time course of the reaction. In the presence of dithionite no ferric subpopulations were observed as the presence of dithionite rapidly (in comparison to the NiR activity) removed ferric protein.

Figure 3B, I see only one green circle. Figure legend corrected.

Table 1, what is the difference between ND and leaving a blank cell?

  • There is one complete set of data for NO binding, but it is sometimes an NO concentration independent rate (s-1) and sometimes an NO concentration depended second order rate constant (M-1s-1). However, for the NOD activity some of the proteins do not show any optical relation, hence not detected. The table has been altered to clarify this.

Round 2

Reviewer 1 Report

The authors indeed have satisfactorily answered most of my comments. 

Although I consider the manuscript worth being published in the present form, I mus slightly disagree with authors on one point, namely on NO dissociation.

I do believe that measurements of NO dissociation in the WT and in various mutant might be relevant for this paper, even though I acknowledge that it is not an easy task. In any case, it would feasible following the displacement of NO by a large excess of CO in a protein solution with 0.1 uM NO (experimentally accessible!). The relevance of the determination of NO dissociation is connected to the mechanism the authors describe as a potential functional role fo cytoglobin.  

Reviewer 4 Report

The Authors have replied all my concerns. Congratulations on an excellent piece of work.